# CATALYST trial protocol: a multicentre, open-label, phase II, multiarm trial for an early and accelerated evaluation of the potential treatments for COVID-19 in hospitalised adults

Tonny Veenith,[1,2] Benjamin A. Fisher [iD],[3,4,5,6] Daniel Slade,[3] Anna Rowe,[3,6] Rowena Sharpe,[3] David R. Thickett,[1,7] Tony Whitehouse [iD],[1,2] Matthew Rowland,[8] James Scriven,[9] Dhruv Parekh [iD],[1,2,7] Sarah J. Bowden,[3] Joshua S. Savage [iD],[3] Duncan Richards,[10] Julian Bion [iD],[2] Pamela Kearns,[3,6] Simon Gates,[3] CATALYST Trial Investigators, On behalf of CATALYST investigators

TV and BAF contributed equally. JB, PK and SG contributed equally.

For numbered affiliations see end of article.

**Correspondence to**
Dr Benjamin A. Fisher;
b.fisher@bham.ac.uk

## ABSTRACT

**Introduction** Severe SARS-CoV-2 infection is associated with a dysregulated immune response. Inflammatory monocytes and macrophages are crucial, promoting injurious, proinflammatory sequelae. Immunomodulation is, therefore, an attractive therapeutic strategy and we sought to test licensed and novel candidate drugs.

**Methods and analysis** The CATALYST trial is a multiarm, open-label, multicentre, phase II platform trial designed to identify candidate novel treatments to improve outcomes of patients hospitalised with COVID-19 compared with usual care. Treatments with evidence of biomarker improvements will be put forward for larger-scale testing by current national phase III platform trials. Hospitalised patients >16 years with a clinical picture strongly suggestive of SARS-CoV-2 pneumonia (confirmed by chest X-ray or CT scan, with or without a positive reverse transcription PCR assay) and a C reactive protein (CRP) ≥40 mg/L are eligible. The primary outcome measure is CRP, measured serially from admission to day 14, hospital discharge or death. Secondary outcomes include the WHO Clinical Progression Improvement Scale as a principal efficacy assessment.

**Ethics and dissemination** The protocol was approved by the East Midlands-Nottingham 2 Research Ethics Committee (20/EM/0115) and given urgent public health status; initial approval was received on 5 May 2020, current protocol version (V.6.0) approval on 12 October 2020. The MHRA also approved all protocol versions. The results of this trial will be disseminated through national and international presentations and peer-reviewed publications.

**Trial registration numbers** EudraCT2020-001684-89, ISRCTN40580903.

## Strengths and limitations of this study

► CATALYST will provide a rapid readout on the safety and proof of concept of candidate novel treatments.
► CATALYST will enable phase III trial resources to be focused and allocated for agents with a high likelihood of success.
► CATALYST uses Bayesian multilevel models to allow for nesting of repeated measures data, with factors for each individual patient and treatment arm, and allowing for non-linear responses.
► CATALYST is not designed to provide a definitive signal on clinical outcomes.

the International Severe Acute Respiratory and emerging Infections Consortium (ISARIC) partnerships indicate that 20% of patients admitted to the hospital require critical care support, with an overall mortality of 34.5% after hospital admission in the last two waves.[1] Mortality following respiratory support remains over 40% in the second wave.[2] Recovery from severe disease may be associated with long-term health impact.[3] [4] Despite the introduction of vaccination programmes in December 2020, there remains an urgent need to identify agents which prevent progression to critical illness, reduce morality and promote rapid recovery.

SARS-CoV-2 can cause severe pneumonia with diffuse alveolar damage, infiltrating perivascular lymphocytes, disrupted endothelial cell membranes, vascular thrombosis with microangiopathy and occlusion of alveolar capillaries.[5] Subsequent multiple organ failure, is in part, driven by a dysregulated immune response. Inflammatory monocytes

## INTRODUCTION

Since the start of the pandemic, the UK has now passed the milestone of 100 000 deaths due to SARS-CoV-2 virus. Current data from

and macrophages contribute to endothelial damage and microthrombosis and drive cytokine production. The cellular response is characterised by an upregulation of proinflammatory cytokines and chemokines, leading to a host immune response targeted at the virus, damaging host tissues.[6] The severity of the disease is proportional to the cytokine response (e.g. interleukin (IL)-6, interferon gamma inducible protein-10, monocyte chemoattractant protein, macrophage inflammatory protein-1A and tumour necrosis factor (TNF)-α), with critically ill patients exhibiting the highest levels of cytokines and chemokines.[7–9] This dysregulated immunity may be a modifiable pathobiological therapeutic target for preventing COVID-19 progression. Potential mechanisms underlying this immune pathobiology may be targeted with precision using existing licensed and novel drugs. The CATALYST platform, therefore, aims to study these drugs and other novel therapeutic options for rapid assessment of safety, biological signal for efficacy and providing secure underpinning of science prior to large phase III trials. Those with potential efficacy can then be considered for larger-scale testing by national platform trials such as Randomised Evaluation of COVID-19 Therapy (RECOVERY)[10] or Randomised, Embedded, Multifactorial Adaptive Platform trial for Community-Acquired Pneumonia (REMAP-CAP).[11]

## METHODS AND ANALYSIS
### Study design
The CATALYST trial is a multiarm, randomised, open-label phase II clinical trial. Initial candidate drugs included within the multiarm design were namilumab, and infliximab, monoclonal antibodies targeting the proinflammatory cytokines granulocyte-macrophage colony-stimulating factor (GM-CSF) and TNF-α, respectively. A third agent, gemtuzumab-ozogamicin (Mylotarg), was incorporated in the protocol but has

not been prioritised for evaluation at this stage and has not opened to recruitment. Each candidate therapy will be given in addition to usual care and compared with usual care independently (figure 1). Randomisation will be performed by an automated minimisation procedure that attempts to allocate participants in a balanced manner between treatment groups allowing for the stratification variable (ward or intensive care unit [ICU]) and with a 20% random component. The WHO Trial Registration Data Set is attached in online supplemental appendix 2.

New therapies may be introduced sequentially utilising this design. Therapies for evaluation will be based on the scientific rationale prioritised by a scientific advisory board (SAB). This will enable the rapid addition of new treatment arms, while allowing the efficacious assessment and analysis of each of these treatment arms.[12 13] Participants are followed up for at least 28 days postrandomisation. A list of protocol changes is detailed in online supplemental appendix 3.

### Patient and public involvement
From its inception, the CATALYST trial was co-developed with the Critical Care patient and public involvement (PPI) group based in the Surgical Reconstruction and Microbiology Research Centre at University Hospitals Birmingham. The group reviewed and refined the protocol and participant-facing documents and provided input into the design; specifically, their feedback supported a usual care arm without the inclusion of placebo controls given the context of the ongoing pandemic. PPI representatives are members of the trial steering committee (TSC) who will supervise the conduct of the trial conduct, and monitor progress, including recruitment and will support the dissemination of the trial results.

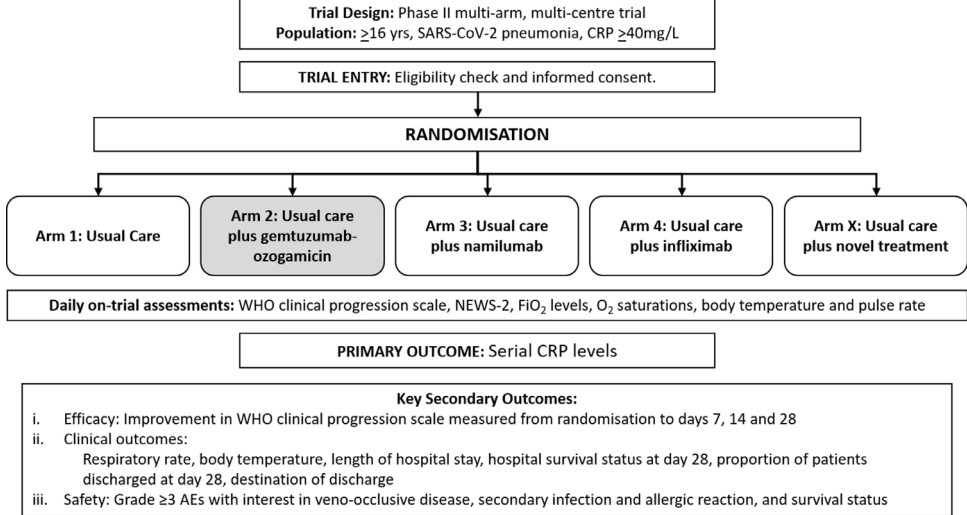

**Figure 1** Catalyst trial schema. The gemtuzumab-ozogamicin arm has been deprioritised and is not open to recruitment. AE, adverse events; CRP, C reactive protein; FiO2, fractional inspired oxygen; NEWS-2, National Early Warning Scale-2.

## Inclusion and exclusion criteria

Hopitalised patients aged 16 years or over, with a clinical picture strongly suggestive of SARS-CoV-2 pneumonia (confirmed by chest X-ray or CT scan, with or without a positive reverse transcription PCR assay), and C reactive protein (CRP) greater than or equal to 40 mg/L are eligible for this study. Exclusion criteria include patient or personal/professional legal representative refusal, planned palliative care, pregnancy or breastfeeding women, women of childbearing potential and non-vasectomised men who are unwilling to use effective contraception for the duration of the trial and throughout the drug-defined post-trial period. Patients with known HIV or chronic hepatitis B or C infection, contraindications to any of the investigational medicinal products (IMPs), receiving concurrent immunosuppression with biological agents, a history of haematopoietic stem cell transplant or solid organ transplant, known hypersensitivity to drug products or excipients, tuberculosis or other severe infections such as (non-SARS-CoV-2) sepsis, abscesses and opportunistic infections requiring treatment, moderate or severe heart failure (NYHA class III/IV), or any other indication or medical history, that in the opinion of the patient's local investigator is unsuitable for trial participation. Patients will not be eligible if they are currently participating in another COVID-19 interventional trial; coenrolment into RECOVERY-Respiratory Support trial is allowed.

## Consent

Patients are identified as per site-established processes. This may include searching central logs of patients admitted with COVID-19, or via prescreening processes already in situ at the site. Each eligible patient who has capacity will be given a patient information sheet (PIS) to read more about the trial (online supplemental appendix 4). Informed consent is requested from patients with capacity by an investigator who has been delegated the responsibility on the delegation log. Where a patient lacks capacity (eg, from severity of illness) informed consent will be sought from the patient's personal legal representative (PerLR). In the event that the PerLR is unavailable, informed consent will be sought from the patient's professional legal representative (ProfLR) according to the requirements of the UK Health Research Authority.[14] Specific PISs for both PerLR and ProfLR can also be emailed to aid this process. If a patient recovers their capacity, they should be reconsented as soon as possible using the standard PIS and informed consent form (ICF). Patient and PerLR forms are also available in Bengali, English, French, Polish, Portuguese, Punjabi, Urdu and Welsh. Online supplemental appendices 4, 5 contain the English version of the patient, ProfLR and PerLR information sheets, and ICFs, respectively.

As soon as the patient is considered eligible the site investigator or delegated team member should enter the patient into the trial by completing the randomisation form on the electronic Remote Data Capture (eRDC) system. This will allocate the participant, as described above, into an open trial arm.

## Interventions

Arm 1: Usual care provided following the current institutional policy for patients with COVID-19. This may vary by site and over time. Following the recommendation from the UK CMO in June 2020, standard care includes dexamethasone treatment.

Arm 3: Usual care combined with namilumab, administered in a single dose (150 mg) on day one infused intravenously over 1 hour. Namilumab is an anti-GM-CSF monoclonal antibody with a good safety profile up to phase II studies in rheumatoid arthritis (RA) and axial spondyloarthropathy with over 360 individuals dosed in total.[15–18] The objective of namilumab therapy in COVID-19 is to inhibit inflammatory monocyte/macrophage activation and their trafficking to the lungs so as to reduce the aberrant immunopathology.

Arm 4: Usual care combined with infliximab, administered in a single dose of 5 mg/Kg diluted in 250 mL of 0.9% saline on day one and infused over a 2-hour period. Infliximab is a widely available anti-TNF alpha monoclonal antibody licensed for the treatment of a number of diseases including RA. TNF is a key proinflammatory cytokine produced by macrophages implicated in a number of processes contributing to early lung pathology.[19]

No dose modification for namilumab or infliximab is permitted. Although there are no requirements for premedication, patients may receive premedication or treatment with antihistamines and paracetamol at local discretion to prevent or treat mild-to-moderate infusion reactions due to namilumab or infliximab administration.

Initially, gemtuzumab-ozogamicin, an antibody-drug conjugate approved for induction therapy of acute myeloid leukaemia was included for investigation in the trial (arm 2), however, prioritisation discussions in the government committee overseeing the COVID-19 phase II studies, advised that this arm should be suspended without recruitment, in favour of continuing with namilumab and infliximab arms.

## Concomitant medication

On admission to hospital, and in accordance with usual care, concomitant medication should be reviewed for contraindications to the IMPs. Where not contraindicated, concomitant medication may be given as medically indicated and in line with the summary of product characteristics or investigational brochure as applicable for that IMP. There are no known contraindications for namilumab. For patients randomised to arm 4, usual care and infliximab, the use of anakinra, abatacept and tocilizumab is not recommended.

## Trial outcomes

The primary outcome is CRP concentration over time, where a sequential reduction in one of the interventional arms as compared with usual care is considered indicative that this may be a clinically effective treatment suitable for testing in phase III clinical trials.

The secondary outcomes are aligned with Core Outcome Measures in Effectiveness Trials' initiatives.[20 21] The principal clinical efficacy measure is the WHO Clinical Progression Improvement Scale measured daily for 28 days on a 1–10 scale; level 0 (no viral load detected) will not be assessed over the course of this study (table 1). Other clinical measures assessed until day 14, discharge or death, include the ratio of the oxygen saturation to fractional inspired oxygen concentration ($SpO_2/FiO_2$), respiratory rate, body temperature and the National Early Warning Scale 2. Assessments assessed until day 28 include length of hospital stay, hospital survival status at day 28, the proportion of patients discharged at day 28 and the destination of discharge. Routine laboratory measurements at baseline, days 3, 5, 7, 9 and 14 include lymphocyte count, neutrophil count, neutrophil: lymphocyte ratio, ferritin, D-dimer and lactate dehydrogenase. Safety measures as defined by adverse events (AEs) and as recorded by Common Terminology Criteria for Adverse Events (CTCAE), V.4.03[22] are those of grade≥3, secondary infection and allergic reaction, and survival status. The schedule of events is shown in table 2 with additional IMP-specific schedules in online supplemental appendix 6.

If the patient is discharged from the hospital before their next scheduled visit, the participant should be provided with the WHO clinical improvement scale diary and the visits on days 7, 14 and 28 should take place by telephone (if it is not possible to see the patient). If the visit is via telephone this will include an AE review and will collect data on WHO clinical improvement scale assessment.

In a subset of the patients (those admitted to University Hospitals Birmingham National Health Service [NHS] Trust and University of Oxford NHS Trust) optional samples consisting of whole blood (for RNA, DNA and also cellular assessments, preserved in Cytodelics buffer), peripheral blood mononuclear cells and plasma, will be obtained on days 1, 3 and 9 (or day of discharge if earlier) and will broadly follow the ISARIC protocol.[23] All samples will be collected in accordance with national regulations and requirements, including standard operating procedures for logistics and infrastructure. Samples will be taken in appropriately licensed premises, stored and transported per the Human Tissue Authority guidelines and NHS trust policies.

## Statistical analysis plan

The primary outcome data will consist of a sequence through time of readings of each patient's CRP. These will be modelled using Bayesian multilevel models (also known as hierarchical or mixed effects) that allow for nesting of the repeated measures data within patient, and

| Table 1 | WHO Clinical Progression Scale | |
|---|---|---|
| **Patient state** | **Descriptor** | **Score** |
| Uninfected | Uninfected; no viral RNA detected | 0 |
| Ambulatory | Asymptomatic; viral RNA detected | 1 |
| | Symptomatic; independent | 2 |
| | Symptomatic; assistance needed | 3 |
| Hospitalised; mild disease | Hospitalised; no oxygen therapy | 4 |
| | Hospitalised; oxygen by mask or nasal prongs | 5 |
| Hospitalised; severe disease | Hospitalised; oxygen by NIV or high flow | 6 |
| | Intubated and mechanical ventilation, $pO_2/FiO_2>150$ or $SpO_2/FiO_2>200$ | 7 |
| | Mechanical ventilation $pO_2/FiO_2<150$ ($SpO_2/FiO_2<200$) or vasopressors | 8 |
| | Mechanical ventilation $pO_2/FiO_2<150$ ($SpO_2/FiO_2<200$) and vasopressors, dialysis or ECMO | 9 |
| Death | Dead | 10 |

Adapted from reference 29.

Footnotes for use in CATALYST. (1) If $pO_2$ not available then use the $SpO_2/FiO_2$ ratio instead. (2). For $pO_2$ measurements in kPa, use an online calculator, for example, https://www.msdmanuals.com/en-gb/medical-calculators/PaO2_FiO2Ratio.htm to calculate a $pO_2/FiO_2$ ratio equivalent to that obtained with $pO_2$ measured in mm Hg, or else consider an equivalent ratio to 200, when dividing $pO_2$ in kPa by $FiO_2$, is 26.7, and an equivalent to 150 is 20. (3). If medically fit for discharge, record status as for ambulatory patient. (4). Asymptomatic implies a return to baseline symptomatic state that is, no fever, and no cough, shortness of breath, confusion, myalgia, diarrhoea, fatigue or weakness above what the participant would have experienced on a daily basis before their COVID-19 episode. (5). Symptomatic but independent, implies that the participant has some of the additional symptoms as above, but needs no additional help with activities of daily living above what they required prior to their COVID-19 episode. (6). Symptomatic but needs assistance, implies that in addition to having symptoms as above, they require help with activities of daily living that is, bathing/showering, personal hygiene and combing of hair, dressing, toileting, mobility/transferring and self-feeding, above what they required on a daily basis prior to their COVID-19 episode. (7). Score 0 (uninfected: no viral RNA detected) is not being assessed as part of CATALYST.
ECMO, extracorporeal membrane oxygenation; NIV, non-invasive ventilation; $pO_2$, partial pressure of oxygen; SpO2/FiO2, oxygen saturation to fractional inspired oxygen concentration.

allowing for non-linear responses. Specifically, posterior probabilities for the treatment/time interaction covariates will be used to conduct decision making. Data will be analysed for each intervention arm against the control group, including in each analysis, only participants who

**Table 2** CATALYST schedule of events

| | Baseline | Day 1 | Day 2 | Day 3 | Day 4 | Day 5 | Day 6 | Day 7 | Day 8 | Day 9 | Day 10 | Day 11 | Day 12 | Day 13 | Day 14 | Day 15 – Day 27 * | Day 28† |
|---|---|---|---|---|---|---|---|---|---|---|---|---|---|---|---|---|---|
| Eligibility assessment | x | | | | | | | | | | | | | | | | |
| Consent | x | | | | | | | | | | | | | | | | |
| Weight/ height (estimated; BSA calculation) | x | | | | | | | | | | | | | | | | |
| Demographics‡ | x | | | | | | | | | | | | | | | | |
| Pregnancy test (females only) | x | | | | | | | | | | | | | | | | |
| Frailty Score, Comorbidity assessment | x | | | | | | | | | | | | | | | | |
| Review of medical history | x | | | | | | | | | | | | | | | | |
| Randomisation | x | | | | | | | | | | | | | | | | |
| WHO clinical progression scale | x | x | x | x | x | x | x | x | x | x | x | x | x | x | x | x | x |
| Routine blood tests § | x | (x) | (x) | x | (x) | x | (x) | x | {x} | x | (x) | (x) | (x) | (x) | x | | |
| Research Samples¶ optional—see section 8 | x | | x | x | | | | | | x*** | | | | | | | |
| National Early Warning Score-2 | x | x | x | x | x | x | x | x | x | x | x | x | x | x | x | | |
| FiO2 levels, O2 saturations, respiratory rate†† | x | x | x | x | x | x | x | x | x | x | x | x | x | x | x | | |
| Body temperature, pulse rate | x | x | x | x | x | x | x | x | x | x | x | x | x | x | x | | |
| Adverse event review | | | | | | | | x | | | | | | | x | | x |
| Concomitant medication review | x | | | | | | | | | | | | | | | | |

Continued

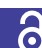

**Table 2** Continued

| | Baseline | Day 1 | Day 2 | Day 3 | Day 4 | Day 5 | Day 6 | Day 7 | Day 8 | Day 9 | Day 10 | Day 11 | Day 12 | Day 13 | Day 14 | Day 15 – Day 27 * | Day 28† |
|---|---|---|---|---|---|---|---|---|---|---|---|---|---|---|---|---|---|
| IMP Administration See additional schedules in online supplemental appendix 6 | x | | | | | | | | | | | | | | | | |

Notes: If the patient is discharged from the hospital before their next scheduled visit, the participant should be provided with the WHO clinical improvement scale diary and the visits on days 7, 14 and 28 should take place by telephone (if it is not possible to see the patient). If the visit is via telephone please perform an adverse event review including asking about any hospitalisations and a WHO clinical improvement scale assessment.

On day 1, tests and interventions should be recorded predose (if randomised to interventional arms).

*Information to be collected at day 28 if patient discharged. A collection tool will be available.

†Information on serious adverse events will be collected until 28 days after the last IMP administration, which may be after this time point.

‡To include age, sex and smoking status (if known).

§Full blood count (WCC, platelets, lymphocytes, neutrophils, monocytes, eosinophils, haemoglobin), D-dimer, C reactive protein, ferritin, lactate dehydrogenase, liver function test (alanine aminotransferase or aspartate aminotransferase, bilirubin, alkaline phosphatase, albumin), urea and electrolytes (urea, creatinine, sodium and potassium) NB. On day of IMP administration, this should be taken pre (up to 24 hours earlier). (x)—not mandatory for usual care but if undertaken for clinical or safety reasons, data will be collected.

¶The sample substudy is only open at specific sites. Samples can be taken±24 hours of day 1. On day 3, samples can be taken up to 24 hours before and up to 48 hours after the visit. Day 9 samples can be taken up to 24 hours before and up to 48 hours after the visit or at discharge if earlier. Please note the samples should be taken before IMP administration on day 1 where possible.

**Or on day of discharge if earlier.

††O₂ and saturation levels will be recorded twice daily.

BSA, body surface area; FiO₂, fractional inspired oxygen concentration; IMP, Investigational Medicinal Product; NB, nota bene; O₂, oxygen; WCC, white cell count.

were eligible for that comparison. The model will be adjusted for age and care status at recruitment (ward or ICU).

At the specified decision points, with interim analysis at n=20 and n=40 and a final analysis at n=60 per arm, CRP data will be considered in the context of the emerging safety data to make a recommendation as outlined below:

► If there is strong evidence of an additional inflammatory effect (CRP) and a satisfactory safety profile consider progression to clinical endpoint evaluation whether in this trial or in another one;

► Terminate arm and do not proceed (based on lack of evidence of an additional biological effect or of an unfavourable safety signal).

Success will be declared if there is a 90% probability that the intervention arm is better than usual care in reducing CRP. Futility is defined as less than 50% probability of the intervention being better than usual care. However, given the large number of agents being investigated in various phase II trials, the size of effect and the totality of data will be reviewed before recommending adoption by a phase III platform. In the event of a successful treatment being identified and the effect size being large, consideration may be given to continuing the arm within CATALYST to study clinical efficacy (based on the WHO scale), if this were deemed to be a more efficient path than translation to a phase III platform. More information, including the operating characteristics based on a simpler analysis of the area under the curve for sequential CRP data are included in online supplemental appendix 7.

New arms will be added as new interventions become available. Each intervention will be compared with temporally relevant usual care controls, using only those patients for whom that intervention was a randomisation option. A detailed secondary outcome measure analysis can be found in a predefined statistical analysis plan (online supplemental appendix 8). The trial statisticians will not be blinded. Exploratory subgroup analyses will be conducted to ascertain the effect of treatment on the primary outcome measure within care status strata (ward or ICU) and disease severity (WHO score <6 or ≥6). Analyses will be conducted as per the primary outcome measure with any inference based on the treatment/time interaction covariate included in the model formulation. Efficacy measurements will be performed primarily on a modified intention to treat population that will include all patients who receive treatment and have at least a baseline and one post-treatment measurement. Missing data will not be imputed. The safety population will include all patients in the usual care arm and all patients who receive a trial intervention in the active arms.

## AEs reporting and analysis

The collection and reporting of AEs will be in accordance with the Research Governance Framework for Health and Social Care and the requirements of the National Research Ethics Service. Definitions of different types of AEs are listed in online supplemental appendix 9. The reporting period for AEs will be between the date of commencement of protocol-defined treatment until day 28. The investigator should assess the seriousness and causality (relatedness) of all AEs experienced by the patient (this should be documented in the source data) with reference to the protocol. Abnormal laboratory findings will only be reported if they satisfy one of the following: (1) events which are grade 3 or above; (2) events which result in the early discontinuation of trial treatment (if applicable to the research arm) or (3) events which result in a dose modification or dose interruption (if applicable to the research arm). Pre-existing conditions and pre-existing abnormal laboratory findings will only be reported if the condition worsens by at least one CTCAE grade. Hospitalisations for preplanned elective procedures, unless the condition worsens, will not be reported as Serious AEs.

## Data management

Data will be collected via a set of forms capturing details of eligibility, baseline characteristics, treatment and outcome details created using FORMAP case report form design software developed by Birmingham Clinical Trials Unit at the University of Birmingham. This trial will use an eRDC system, with the exception of SAE reporting and notification of pregnancy; these are both paper based. All trial records must be archived and securely retained for at least 25 years. No documents will be destroyed without prior approval from the sponsor, via the central CATALYST Trial Office. On-site monitoring will be carried out as required following a risk assessment and as documented in the quality management plan. Any monitoring activities will be reported to the central CATALYST Trial Office and any issues noted will be followed up to resolution. CATALYST will also be centrally monitored, which may trigger additional on-site monitoring. The trial management group (TMG) and authors will have access to the final dataset. Further information regarding data management is provided in the study protocol.

## Trial organisation structure

The University of Birmingham will act as single sponsor for this multi-centre study: Support Group, Aston Webb Building, Room 119, Birmingham, B15 2TT. Email: researchgovernance@contacts.bham.ac.uk). The trial is being conducted under the auspices of the Cancer Research UK Clinical Trials Unit, University of Birmingham, in close partnership with the National Institute for Health Research Biomedical Research Centres (BRC) at the Universities of Birmingham and Oxford, University College London (UCL) and Imperial College London.

Given the combination of a novel disease, a range of novel potential therapies and a pandemic setting, a multidisciplinary collaboration was essential, bringing together experts in acute, respiratory and intensive care medicine, inflammation, oncology, data sciences, trials methodology and statistics. The Birmingham Acute Care

Research group provides a single point of reference for the acute specialties.[24]

The TMG is responsible for the day-to-day running and management of the trial. Members include the chief investigator, deputy chief investigator, coinvestigators, trial statisticians, trial management team leader and trial coordinator. The TMG reports to the TSC.

The TSC provides oversight and governance. Members include independent clinicians and patient advocates. The TSC supervises the conduct of the trial, monitoring progress including recruitment, data completeness, lost to follow-up and deviations from the protocol. They will make recommendations about conduct and continuation of the trial.

The independent data monitoring committee (DMC) includes clinicians and a statistician who will review unblinded data analyses to advise the TSC on whether the trial data (and results from other relevant research), justifies the continuing recruitment of further patients. The DMC will operate in accordance with a trial-specific charter based on the template created by the Damocles Group. The DMC will review the trial data 3 monthly during the recruitment and while patients remain on treatment. These may occur more frequently if the DMC deem necessary or for interventions which have not previously been administered to patients in this specific setting.

The SAB makes recommendations on prioritising interventions and aspects of methodology such as coenrolment in order to harmonise trial activities with other research platforms. The SAB includes multistakeholder representatives (detailed in online supplemental appendix 1) including, the collaborating centres and the associated BRC[25] in Birmingham,[26] Oxford[27] and UCL.[28]

### Confidentiality statement

Confidential trial data will be stored in accordance with the General Data Protection Regulation 2018. As specified in the PIS and with the patient's consent, patients will be identified using only their date of birth and unique trial ID number.

### Trial status

Recruitment for the trial opened in May 2020 with the namilumab and infliximab arms. Although included in the protocol, the gemtuzumab-ozogamicin arm has not opened to recruitment.

### DISCUSSION

CATALYST is a nimble, accelerated, open-label, targeted phase II proof-of-principle multiarm trial permitting efficient evaluation of repurposed and/or novel drugs to modify the disease progression of COVID-19 in patients admitted to wards and ICU. CATALYST aims to determine suitability of a proposed new treatment for evaluation in phase III national platform trials. CATALYST has an adaptive platform design, aiming to translate laboratory-based research to patients with SARS-CoV-2 infection without delay.

This trial is designed with clinical pressures caused by the pandemic in mind. Outcomes are easy to record, location-independent and applicable across the spectrum of illness severity. We have also prioritised serial continuous measures over discontinuous or ordinal metrics, as these allow for greater statistical efficiency and thus smaller sample sizes. The primary outcome chosen provides a rapid, biologically driven efficacy signal to allow early 'go/no-go' decisions. While the WHO has adopted a consensus-based set of core outcome measures for studies of SARS-CoV-2 infection,[29] our study aim was to develop a trial with a smaller sample size to provide earlier signals of potential efficacy for multiple investigational agents, allowing selection of the most promising to be taken forward by larger platforms with clinical outcomes.

We initially considered $SpO_2/FiO_2$ as a suitable primary outcome for CATALYST. This is an indicator of the severity, progression or remission of acute lung injury.[30] However, as real-world data emerged, we found the relationship between the ratio and outcome was complex and was also susceptible to measurement error in patients receiving ward-based forms of respiratory support. Furthermore, its prognostic utility is compromised if the inspired oxygen concentration is not rapidly adjusted to the patients' needs by the attending staff, which is often delayed in a pandemic. Other shortfalls included high variability between observations, the impact on the measure of switching the mode of oxygen delivery, and microthrombotic events altering $SpO_2/FiO_2$ through alteration of perfusion.[31] We, therefore, evaluated whether CRP would be a better primary outcome.

CRP is produced as a homopentameric protein, termed native CRP, which can irreversibly dissociate at sites of inflammation, tissue damage and infection into five separate monomers, termed monomeric CRP.[32] CRP levels are widely used as a marker of infection or inflammation; however, evidence suggests that CRP plays an active role in the inflammatory process.[33–38]

During the first wave, we modelled data on CRP over time in COVID-19 patients finding that it performed better than $SpO_2/FiO_2$. This is consistent with published data indicating that baseline and peak CRP, median CRP over time, slope of CRP rise over 7 days, and rapid rise during early disease are all associated with outcome in hospitalised patients with COVID-19.[39–42] CRP trends over time tend to have greater predictive power, with change in CRP levels performing better at predicting respiratory failure and subsequent intubation than baseline CRP or respiratory rate-oxygenation index.[40] In addition, compared with those that die, patients who survive have lower peak CRP levels and earlier reductions.[41] Notably, CRP at baseline also correlates with CT grading of lung involvement.[43] A recent large study found that CRP ≥40 mg/L, a key entry criteria for our study, was the optimal CRP cut-off for predicting mortality on hospital admission.[44]

The recent success of dexamethasone in the treatment of COVID-19 reinforces the relevance of inflammatory pathology to clinical outcomes.[45] Methylprednisolone has been associated with CRP reduction over 7 days and improved clinical outcomes, in a strategy that continued this steroid until a target CRP, or the ratio of arterial oxygen partial pressure to fractional inspired oxygen ($PiO_2/FiO_2$) threshold, was reached.[46] Although IL-6 blockade has a direct effect on CRP production, which might obscure a relationship between CRP trends and outcome, small studies have suggested a differential effect on CRP decline between clinical responders and non-responders.[47 48]

While CRP may not show complete concordance with clinical outcomes, we argue that if dysregulated inflammation is indeed a key pathogenic driver in severe COVID-19, an immunomodulating drug capable of ameliorating those outcomes is also likely to show early improvement of CRP. Conversely, an immunomodulating drug unable to influence CRP is a less promising candidate to investigate in large phase III trials. Limitations of CRP, however, include lower utility for candidate therapeutics whose mechanism of action is not immunomodulation, and the diminished ability to assess drugs that directly target IL-6, due to the direct pharmacodynamic effects on CRP. This study was designed before use of IL-6 blockade outside of trials, and if tocilizumab were to be widely adopted, adaptive modification of the current trial design may be required to account for this.

In conclusion, the major strength of CATALYST is its ability to provide a rapid readout on safety, and proof-of-efficacy enabling phase III trial resources to be focused and allocated for drugs with a high likelihood of success.[45 49] This will reduce the time lag in translating early phase drugs into effective therapeutics for COVID-19.

## ETHICS AND DISSEMINATION

The trial will be performed in accordance with the recommendations guiding physicians in biomedical research involving human subjects, adopted by the 18th World Medical Association General Assembly, Helsinki, Finland and stated in the respective participating countries laws governing human research, and Good Clinical Practice. The initial protocol was approved by East Midlands-Nottingham 2 Research Ethics Committee, (REC Ref: 20/EM/0115) on 5 May 2020, with subsequent amendments approved on 28 May 2020 (addition of namilumab and infliximab), 12 June 2020 (inclusion change-suspected COVID-19), 20 June 2020 (following dexamethasone as standard of care use), 12 October 2020 (change of primary outcome to CRP). The MHRA has given its approval of all protocol versions; current version in use is 6.0.

A meeting will be held after the end of the trial to allow discussion of the main results among the collaborators prior to publication. Results of the primary and secondary endpoints will be submitted for publication in peer-reviewed journals. Manuscripts will be prepared by the TMG and authorship will be determined by mutual agreement.

**Author affiliations**
[1]Birmingham Acute Care Research Group, Institute of Inflammation and Ageing, College of Medical and Dental Sciences, University of Birmingham, Birmingham, UK
[2]Department of Critical Care Medicine, University Hospitals Birmingham NHS Foundation Trust, Birmingham, UK
[3]Cancer Research UK Clinical Trials Unit, Institute of Cancer and Genomic Sciences, University of Birmingham, Birmingham, UK
[4]Rheumatology Research Group, Institute of Inflammation and Ageing, College of Medical and Dental Sciences, University of Birmingham, Birmingham, UK
[5]Department of Rheumatology, University Hospitals Birmingham NHS Foundation Trust, Birmingham, UK
[6]National Institute for Health Research (NIHR) Birmingham Biomedical Research Centre, University Hospitals Birmingham NHS Foundation Trust, Birmingham, UK
[7]Department of Respiratory Medicine, University Hospitals Birmingham NHS Foundation Trust, Birmingham, UK
[8]Nuffield Department of Clinical Neurosciences, John Radcliffe Hospital, University of Oxford, Oxford, UK
[9]Department of Infectious Diseases, University Hospitals Birmingham NHS Foundation Trust, Birmingham, UK
[10]Oxford Clinical Trials Research Unit, Botnar Research Centre, University of Oxford, Oxford, UK

**Acknowledgements** We thank staff from the CRCTU, University of Birmingham including Dr Siân Lax for contributions to the paper.

**Collaborators** CATALYST Trial Investigators.

**Contributors** Study conception: TV, BF, DS, AR, RS, DT, TW, MR, DP, DR, JB, PK and SG. Study design: TV, BF, DS, AR, RS, DT, TW, MR, JS, DP, SB, JS, DR, JB, PK, SG. DS is the Trial Biostatistician and SG is the Senior Trial Biostatistician, both were responsible for developing the statistical plan.

**Funding** This trial is supported by the Medical Research Council (MRC) grant number MC_PC_20007. SG is supported by a Senior Investigator Award from the National Institute of Health Research. Staff at the CRCTU are supported by core funding grants from Cancer Research UK (C22436/A25354), the NIHR Biomedical Research Centre (BRC-1215-20009), The Kennedy Trust for Rheumatology Research as part of the Arthritis-Trials Acceleration Programme (KENN161704), and Innovate UK as part of the Midlands-Wales Advanced Therapy Treatment Centres (104232). This paper presents independent research supported by the NIHR Birmingham Biomedical Research Centres at the University Hospitals Birmingham NHS Foundation Trust and the University of Birmingham, as well as Oxford and University College London Hospitals Biomedical Research Centres. Namilumab is being provided free of charge by Izana Bioscience, Oxford, UK (now part of Roivant). Infliximab is being provided free of charge by Celltrion.

**Disclaimer** The views expressed are those of the authors and not necessarily those of the NHS, the NIHR or the Department of Health and Social Care. Neither the sponsor or funders had any role in trial design, data collection, data analysis, data interpretation or writing of the report. The corresponding author had full access to all the data in the trial and had final responsibility for the decision to submit for publication.

**Competing interests** BF has undertaken consultancy for Novartis, BMS, Servier, Galapagos and Janssen; MR is currently undertaking a Senior Clinical Fellowship financed by Roche; PK has undertaken consultancy for BMS, AstraZeneca, and AbbVie; all are unrelated to this trial. All other authors declare no competing interests.

**Patient consent for publication** Not applicable.

**Provenance and peer review** Not commissioned; externally peer reviewed.

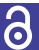

terminology, drug names and drug dosages), and is not responsible for any error and/or omissions arising from translation and adaptation or otherwise.

**ORCID iDs**
Benjamin A. Fisher http://orcid.org/0000-0003-4631-549X
Tony Whitehouse http://orcid.org/0000-0002-4387-3421
Dhruv Parekh http://orcid.org/0000-0002-1508-8362
Joshua S. Savage http://orcid.org/0000-0003-0599-0245
Julian Bion http://orcid.org/0000-0003-0344-5403

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
