## [Reviewer comments · BMJ Open]

ARTICLE DETAILS

TITLE (PROVISIONAL)	CATALYST trial protocol: A multicentre, open-label, phase II, multi-arm trial for an early and accelerated evaluation of the potential treatments for COVID-19 in hospitalised adults
AUTHORS	Veenith, Tonny; Fisher, Benjamin; Slade, Daniel; Rowe, Anna; Sharpe, Rowena; Thickett, David; Whitehouse, Tony; Rowland, Matthew; Scriven, James; Parekh, Dhruv; Bowden, Sarah; Savage, Joshua; Richards, Duncan; Bion, Julian; Kearns, Pamela; Gates, Simon

VERSION 1 – REVIEW

REVIEWER	Ben Carter King's College London, Biostatistics & Health Informatics
REVIEW RETURNED	27-Apr-2021

GENERAL COMMENTS	I congratulate the experienced team for submitting this protocol for their MAMS trial. Whilst I offer a number of comments, please see this in the light of improving the clarity of the manuscript. 1) Primary outcome. Can this be defined in a clearer manner. Is CRP serial, and how is this combined with death or discharge. The assumption is that discharge is due to treatment success - similarly with mortality. 2) Nosocomial infection. Were these excluded ? 3) CRP - The following reference(s) may defend the use of CRP as well as the threshold of ≥ 40 Stringer et al, The role of C-reactive protein as a prognostic marker in COVID-19. International Journal of Epidemiology, 2021, 1–10 doi: 10.1093/ije/dyab012 Hewitt J, et al. The effect of frailty on survival in patients with COVID-19 (COPE): a multicentre, European, observational cohort study. Lancet Public Health. 2020 Jun 30:S2468-2667(20)30146-8 4) Arms: Arm 2 is not mentioned. Maybe clarify 5) Arm 1: Usual care. Clarify this may vary across site and over time 6) SAP - Will a SAP be drafted prior to analysis and will the analysts be blinded 7) Secondary outcomes. Whilst these are implied- please show these in a time-point by outcomes matrix 8) How did you measure adherence ? did this vary by arm?
--

	9) Population under investigation - to confirm 10) Missing data - please confirm approach taken 11) Analysis, is only age and sex adjusted in the analysis or any other covariates? 12) Analysis - How will the secondary outcomes be analysed and reported eg with a 95% CI without p-value? (more details to be included in the SAP?) 13) Was this carried out within a CTU and what standards (eg SOPs) did this follow 14) Subgroups and sensitivity analysis ? (Please confirm these are/are not being undertaken) 15) How was the randomization sequence generated 16) How were the patients approached, screened, baseline data collected then randomised 17) Outcome assessors - were all outcomes objective ?
--	---

REVIEWER	Sean Ewings University of Southampton, Health Sciences
REVIEW RETURNED	06-May-2021

GENERAL COMMENTS	Thank you; this was a well written paper that was easy to follow, and represents an important study. I have only a few requests for clarifications/additions:  - It would be helpful to include some information on how the randomisation sequence is produced and implemented, and in particular stating if allocation concealment is achieved or not. (I note that the PIS says allocation is by a computer, but it would be helpful to include this in the main article too.) - Could you clarify if "comparisons will be performed temporally" (Statistical analysis plan section) means you are only using control participants randomised within the time period a candidate treatment was open to recruitment? - It may help to present the study observations procedure as a table (per SPIRIT statement recommendations), though I acknowledge these are given in the Trial Outcomes section. - It was noted that level 0 of the WHO scale could not be assessed, but I also wondered if that would also be true of level 1 (asymptomatic but with viral load detected) - otherwise, I'm not clear what the difference is between these levels in terms of your ability to measure it. - Supplementary appendix 4 makes reference to table 5, which is not contained in the article or appendices. The information it is said to contain seems to be in the captions of Tables 2A and 2B though. - It would be useful to know what prior distributions were used for the simulations presented in appendix 4.
---

VERSION 1 – AUTHOR RESPONSE

Reviewer: 1

Dr. Ben Carter, King's College London

Comments to the Author:

I congratulate the experienced team for submitting this protocol for their MAMS trial. Whilst I offer a number of comments, please see this in the light of improving the clarity of the manuscript.

We thank the reviewer for their comments.

1) Primary outcome. Can this be defined in a clearer manner. Is CRP serial, and how is this combined with death or discharge. The assumption is that discharge is due to treatment success - similarly with mortality.

CRP is modelled over time, as described in the Statistical analysis section, using serial readings. In order to further clarify this we have added 'over time' under the Trial Outcomes section. CRP is used in this instance as a biomarker and is not combined with death or discharge data.

2) Nosocomial infection. Were these excluded ?

Nosocomial infections were not excluded unless they met the exclusion criteria listed in the manuscript: 'tuberculosis or other severe infections such as (non-SARS-CoV-2) sepsis, abscesses, and opportunistic infections requiring treatment'

3) CRP - The following reference(s) may defend the use of CRP as well as the threshold of ≥ 40

Stringer et al, The role of C-reactive protein as a prognostic marker in COVID-19. International Journal of Epidemiology, 2021, 1–10 doi: 10.1093/ije/dyab012

Hewitt J, et al. The effect of frailty on survival in patients with COVID-19 (COPE): a multicentre, European, observational cohort study. Lancet Public Health. 2020 Jun 30:S2468-2667(20)30146-8

We thank the reviewer for pointing out this recent study of relevance our entry criteria and primary outcome. We have included the first reference, as most relevant to our trial, and added the following to the discussion: 'A recent large study found that CRP ≥ 40 mg/L, a key entry criteria for our study, was the optimal CRP cut-off for predicting mortality on hospital admission.'

4) Arms: Arm 2 is not mentioned. Maybe clarify

The following text was already included in the intervention section: 'Initially, gemtuzumab-ozogamicin, an antibody-drug conjugate approved for induction therapy of acute myeloid leukaemia was included for investigation in the trial (Arm 2), however, prioritisation discussions in the Government Committee overseeing the COVID-19 phase II studies, advised that this arm should be suspended without recruitment, in favour of continuing with namilumab and infliximab arms.'

5) Arm 1: Usual care. Clarify this may vary across site and over time

We have added the following text to the intervention section: 'This may vary by site and over time.'

6) SAP - Will a SAP be drafted prior to analysis and will the analysts be blinded

A SAP has been finalised and we have now added this as a supplemental appendix. We have added 'predefined' to the mention of the SAP in the paper. We have also added the following; 'The trial statisticians will not be blinded.'

7) Secondary outcomes. Whilst these are implied- please show these in a time-point by outcomes matrix

Thank you. We have now added the schedule of events as Table 2, with addition tables in a supplementary appendix.

8) How did you measure adherence ? did this vary by arm?

The interventions that were added to usual care were given at a single time point. Patients in the interventional arms who did not receive the study intervention were not included in the efficacy

analysis. We have added the following statement: 'Efficacy measurements will be performed primarily on a modified intention to treat population that will include all patients who receive treatment and have at least a baseline and one post-treatment measurement.'

9) Population under investigation - to confirm

We thank the reviewer for noting this omission. In addition to the statement added in response to point 8. We have also added: 'The safety population will include all patients in the usual care arm and all patients who receive a trial intervention in the active arms.'

10) Missing data - please confirm approach taken

We have added the following statement to the statistical section: 'Missing data will not be imputed.'

11) Analysis, is only age and sex adjusted in the analysis or any other covariates?

To clarify this point we have added the following: 'The model will be adjusted for age and care status at recruitment (ward or ICU).'

12) Analysis - How will the secondary outcomes be analysed and reported eg with a 95% CI without p-value? (more details to be included in the SAP?)

This is detailed in the SAP that is now included as a Supplementary Appendix

13) Was this carried out within a CTU and what standards (eg SOPs) did this follow

As stated under the section Trial Organisation Structure, 'The trial is being conducted under the auspices of the Cancer Research UK Clinical Trials Unit (CRCTU)'. The CTU SOPs were followed for this study.

14) Subgroups and sensitivity analysis ? (Please confirm these are/are not being undertaken)

As stated: 'Exploratory subgroup analyses will be conducted to ascertain the effect of treatment on the primary outcome measure within care status strata' – we have now added '(ward or ICU) and disease severity (WHO score <6 or ≥6)' to this statement.

15) How was the randomization sequence generated

To increase clarity we have changed the statement to the following: 'Randomisation will be performed by an automated minimisation procedure that attempts to allocate participants in a balanced manner between treatment groups allowing for the stratification variable (ward or ICU) and with a 20% random component.'

16) How were the patients approached, screened, baseline data collected then randomised

This is detailed under consent. We have added the following statement to complete this: 'As soon as the patient is considered eligible the site Investigator or delegated team member should enter the patient into the trial by completing the Randomisation Form on the electronic Remote Data Capture (eRDC) system. This will allocate the participant, as described above, into an open trial arm.'

17) Outcome assessors - were all outcomes objective ?

There were no subjective measures included

Reviewer: 2

Dr. Sean Ewings, University of Southampton Comments to the Author:

Thank you; this was a well written paper that was easy to follow, and represents an important study. I have only a few requests for clarifications/additions:

- It would be helpful to include some information on how the randomisation sequence is produced and implemented, and in particular stating if allocation concealment is achieved or not. (I note that the PIS says allocation is by a computer, but it would be helpful to include this in the main article too.) Thank you. We have amended the randomisation statement as above.

- Could you clarify if "comparisons will be performed temporally" (Statistical analysis plan section) means you are only using control participants randomised within the time period a candidate treatment was open to recruitment?

That is correct. We have changed this statement to the following that we hope is clearer: 'Each intervention will be compared to temporally-relevant usual care controls, using only those patients for whom that intervention was a randomisation option.'

- It may help to present the study observations procedure as a table (per SPIRIT statement recommendations), though I acknowledge these are given in the Trial Outcomes section. We have now added the schedule of events as Table 2, with addition tables in a supplementary appendix.

- It was noted that level 0 of the WHO scale could not be assessed, but I also wondered if that would also be true of level 1 (asymptomatic but with viral load detected) - otherwise, I'm not clear what the difference is between these levels in terms of your ability to measure it. Level 1 indicates an asymptomatic patient that may be viral load positive or negative as repeat PCR assays were not performed.

- Supplementary appendix 4 makes reference to table 5, which is not contained in the article or appendices. The information it is said to contain seems to be in the captions of Tables 2A and 2B though.

Thank you for pointing this out. We have removed the reference to table 5 as the information is contained in the text/captions.

- It would be useful to know what prior distributions were used for the simulations presented in appendix 4.

Thank you for pointing out this omission. We have added the following statement to the supplementary appendix : 'Simulations were performed in Fixed and Adaptive Clinical Trial Simulator (FACTS) software using default non-informative priors.'

VERSION 2 – REVIEW

REVIEWER	Ben Carter King's College London, Biostatistics & Health Informatics
REVIEW RETURNED	07-Sep-2021
GENERAL COMMENTS	Comments fully addressed